# An Insight into the Factors Influencing Specificity of the SUMO System in Plants

**DOI:** 10.3390/plants9121788

**Published:** 2020-12-17

**Authors:** Moumita Srivastava, Ari Sadanandom

**Affiliations:** Department of Biosciences, Durham University, Stockton Road, Durham DH1 3LE, UK; moumita.srivastava@durham.ac.uk

**Keywords:** post-translational modifications, SUMOylation, SUMO proteases, specificity, subcellular localisation

## Abstract

Due to their sessile nature, plants are constantly subjected to various environmental stresses such as drought, salinity, and pathogen infections. Post-translational modifications (PTMs), like SUMOylation, play a vital role in the regulation of plant responses to their environment. The process of SUMOylation typically involves an enzymatic cascade containing the activation, (E1), conjugation (E2), and ligation (E3) of SUMO to a target protein. Additionally, it also requires a class of SUMO proteases that generate mature SUMO from its precursor and cleave it off the target protein, a process termed deSUMOylation. It is now clear that SUMOylation in plants is key to a plethora of adaptive responses. How this is achieved with an extremely limited set of machinery components is still unclear. One possibility is that novel SUMO components are yet to be discovered. However, current knowledge indicates that only a small set of enzymes seem to be responsible for the modification of a large number of SUMO substrates. It is yet unknown where the specificity lies within the SUMO system. Although this seems to be a crucial question in the field of SUMOylation studies, not much is known about the factors that provide specificity. In this review, we highlight the role of the localisation of SUMO components as an important factor that can play a vital role in contributing to the specificity within the process. This will introduce a new facet to our understanding of the mechanisms underlying such a dynamic process.

## 1. Introduction

Post-translational modifications of proteins play a crucial role in many cellular processes through their unique ability to alter rapidly and reversibly the function, stability, and locations of pre-existing proteins. Protein modifications include chemical modifications such as acetylation and phosphorylation and also the conjugation of proteins to target substrates like ubiquitination and SUMOylation. Both ubiquitination and SUMOylation involve the covalent attachment of ubiquitin and SUMO (Small Ubiquitin-like MOdifier), respectively, to the lysine residues on the target protein. Although ubiquitin is generally regarded as targeting a protein for degradation, SUMOylation can rapidly change the overall fate of target proteins by altering their stability or interaction with partner proteins or DNA. SUMOylation affects several important processes in plants, especially during environmental stresses. These may be abiotic stress such as phosphate deficiency, heat, low temperature, and drought [1,2,3,4,5,6,7,8] or biotic stresses like defence responses to pathogen infection [9,10]. SUMOylation has also been reported to affect physiological processes like flowering time, cell growth and development, and nitrogen assimilation, root growth, light induced seedling growth [11,12,13].

SUMO is a small polypeptide roughly 100–115 amino acids in length that was first identified in plants in *Solanum lycopersicum* [14]. SUMOylation is widespread in all classes of eukaryotes, from single-celled yeast *Saccharomyces cerevisiae* to all plants and mammals, including humans. The first eukaryotic SUMO homolog identified, suppressor of mif two 3 (Smt3), was discovered in *Saccharomyces cerevisiae* [15]. Since then, four SUMO proteins have been identified in animals: SUMO1, 2, 3, and 4. SUMO2 and 3 from human and animals share 87–95% protein sequence identity but are only ~50% identical to SUMO1 [15]. The first plant SUMO, T-SUMO, was found as an interacting partner of ethylene-inducing xylanase (EIX) from the plant pathogenic fungus *Trichoderma viride* [16]. The *Arabidopsis* genome encodes eight isoforms, although only SUMO1, 2, 3, and 5 expression has been verified [17,18,19]. SUMO1 and SUMO2 are the most closely related isoforms, sharing 83% amino acid sequence identity. SUMO3 and SUMO5 display 42% and 30% amino acid sequence identity with SUMO1, respectively [20,21]. As the transcripts of only AtSUM1, 2, 3, and 5 have been detected so far, this suggests that the other isoforms could be expressed in condition-, time-, or space-specific manners. The detection of AtSUM1/2 and their conjugates in both cytoplasmic and nuclear compartments implies that both soluble and membrane-associated proteins may be targets of AtSUM1/2 [17]. In addition to Arabidopsis and tomato, SUMO gene families are present in all other plant species [17,22,23]. Among the different plant species studied, tomato, *Populus* sp., and maize have four SUMO isoforms each, whereas grapevine has only three SUMO isoforms. Both rice and maize have two SUMO isoforms each, while only one SUMO isoform has been identified in Sorghum and *Brachypodium* so far [24].

## 2. Mechanism of SUMOylation

The SUMO modification pathway comprises a target-directed, energy-dependent enzymatic cascade composed of three enzymes that ensures the formation of the isopeptide bond with lysine residues of substrates [25]. In order to empower SUMO for enzymatic reactions, a maturation step is involved that utilises specific proteases that expose the conserved C-terminal di-glycine motif (-Gly-Gly) [26,27]. Subsequently, the enzymatic cascade is constituted by a single heterodimeric E1-activating enzyme (Sae1/Sae2), which activates and links SUMO to an internal cysteine residue, forming a thioester bond, consuming ATP in the process [28,29,30]. This is followed by SUMO being transferred to another unique E2-conjugating enzyme, also by the formation of a thioester bond with the active site cysteine. Finally, the charged-E2 (E2~ SUMO thioester) can either transfer SUMO directly to one or more lysines in the target substrate by mono-, multi-, and poly-SUMOylation [31,32,33], or it can be enzymatically stimulated by an E3-ligase enzyme, which enhances the transfer to the substrate by several fold [33,34]. Additionally, since SUMO conjugation is a reversible process, specific cellular SUMO proteases can cleave the isopeptidic bond between the substrate and SUMO, and the free SUMO becomes available for another conjugation cycle [35,36,37] (Figure 1). Thus, SUMO conjugation is a highly dynamic process in which the presence of free SUMO is tightly regulated in the cell by a balance between conjugation and deconjugation [38].

## 3. Components of the SUMO Machinery

The SUMO system involves enzymes responsible for carrying out three different steps: activation, conjugation, and ligation and deconjugation. The first step is brought about by the SUMO activating enzyme E1. The SUMO-E1 in plants occurs as a heteromeric protein complex that catalyses a three-part reaction, the first being that the C-terminus carboxy group of SUMO interacts with ATP to form adenylated SUMO and pyrophosphate. In the next step, the thiol group (S) of an active site cysteine of E1 reacts with the SUMO adenylate, forming a thio-ester bond between the E1 and the SUMO C-terminus and releasing AMP. In the final step, activated SUMO is transferred to an active site cysteine of E2, a conjugation enzyme (in plants called SCE1), forming a SUMO-E2 thioester intermediate that promotes the transfer of SUMO to a lysine residue in the target protein, mediated directly or indirectly by an E3 ligase [28,39]. The E2 SUMO conjugation activity of AtSCE is ATP-dependent [17]. The next set of enzymes are the SUMO E3 ligases that bind to the SUMO E2 conjugating enzyme and promote the transfer of SUMO to substrates, functioning as a bridge or adaptor between the E2 and the substrate [40]. The SUMO ligases identified in Arabidopsis are SIZ1, MMS21, and PIAL1/2.

The noncovalent interactions between SUMO and the target proteins are attributed to a short stretch of hydrophobic amino acids termed a SUMO Interacting Motif (SIM) on target proteins [41] Structural and functional studies determined that a hydrophobic grove surrounded by basic residues in SUMO is crucial for SIM interaction. During conjugation, SUMO paralogue selection can be mediated by SIM-dependent recruitment of targets to SUMO thioester charged E2 and/or SUMO modified E2. [20]. SIMs have been uncovered in a wide range of proteins, including SUMO enzymes, SUMO substrates, SUMO-binding proteins, and SUMO-targeted Ubiquitin Ligases (STUbLs) [42].

Another crucial component of the machinery is the SUMO proteases that are required in two stages in a SUMO cycle, the first being the maturation of SUMO proteins (hydrolase activity) and the second being the removal of SUMO from targets (de-SUMOylation) (isopeptidase activity). SUMOylation is a reversible process in which de-modification involves the SUMO terminal glycine being removed from the lysine residues of the target protein by specific proteases. The mammalian genome encodes six SUMO-specific proteases termed as SENPs, whereas *Arabidopsis* has seven SUMO-specific proteases, the ULPs. [43]. Recently, two new classes of SUMO protease have been identified in animals. Two novel SUMO proteases were identified in mice, termed DeSI1 (deSUMOylating isopeptidase1) and DeSI2, which lacked sequence similarity to ULP enzymes [44]. Orosa et al. (2018) identified eight putative DeSI proteases in Arabidopsis based on sequence similarity to human DeSI1/2 and functionally characterised one protein, DeSI3a [13]. Another SUMO protease recently identified in humans is called USPL1 (ubiquitin-specific protease-like 1) [45]; currently, however, no homologues have been identified in Arabidopsis.

Besides deSUMOylation by SUMO proteases, the SUMOylated proteins have an alternate route to reverse SUMO modifications through SUMO-Targeted Ubiquitin Ligases (STUbLs) [46]. Additional SUMO moieties are added to the target protein to form a polySUMO chain that acts as a binding domain for a group of ubiquitin E3 ligases called SUMO-Targeted Ubiquitin Ligases (STUbLs) [47]. The STUbLs highlight an important cross-talk between the SUMO and the ubiquitin systems because they mediate an unique form of regulated protein turnover stimulated by polySUMO chains [47]. STUbLs bind polySUMO chains noncovalently through their multiple SIMs, which mediate their dimerisation, a function essential for their activity and the eventual polyubiquitination of both SUMO and its attached targets. These polyubiquitinated proteins are then targeted for degradation by the ubiquitin–proteasome system [47]. Arabidopsis genes *Protein Inhibitor of Activated STAT Like (PIAL)1* and *PIAL2* encode SUMO E4 ligase activities responsible for polySUMO chain formation [46]. There are at least six STUbLs identified in Arabidopsis, including two that are conserved throughout eukaryotes and are evolutionary and functionally equivalent to the mammalian and yeast STUbLs Ring Finger Protein (RNF4) and Synthetic lethal genes8 (Slx8) [46].

## 4. Specificity within the SUMO System

The SUMO status of a target protein is crucial for its function as it is likely to affect its interaction with other proteins. SUMOylation is observed to be highly specific as only a certain set of proteins are SUMOylated or deSUMOylated under any particular stress; however, the factors that contribute to target specificity are still not clear. Many theories have been proposed to explain the source of specificity within the system. SUMOylation is a reversible process and instrumental for the regulation and directionality of cellular pathways and signalling cascades. SUMOylation specifically targets a set of proteins in response to different stimulus, and thus, the components of the machinery usually act highly selectively. However, the factors that affect this selectivity are still not clear. Very few enzymes have been identified that mediate SUMOylation and deSUMOylation, and their specificity in vitro seems irregular. Therefore, the question of specificity in the process remains largely unexplained. SUMO targets a wide variety of proteins and no conserved patterns have been observed so far that could explain the specificity emanating from the target proteins themselves. This clearly shows that some other mechanism exists that regulates the specificity of SUMOylation. We propose that the localisation of the SUMO components at the cellular and the organ level decides the proteins that can be SUMOylated at any specific condition to give a strictly defined mechanism. Figure 2 gives an insight into the distribution of identified SUMO components across various cellular compartments.

## 5. Localisation of SUMO Components in the Nucleus

The nucleus is the control centre of a cell as it regulates the integrity of genes and gene expression. There is clear evidence of many SUMO targets in the nucleus in both plant and animal systems. The E1 activating enzymes SAE1/SAE2, in the animal system, are predominantly nuclear but also appear to be associated with filaments of the nuclear pore complex projecting into both the nucleus and the cytoplasm [47,48]. Similarly, in Arabidopsis, the SUMO E1 activating enzyme displays nuclear localisation [49]. The E2 conjugating enzyme SCE1 is localised in the nucleus and cytoplasm [50,51]. The SUMO E3 ligase, SIZ1, consistent with the presence of an NLS on its C-terminus, localises to the nucleus and nuclear speckles [1]. The second known E3 ligase, MMS21/HPY2, predominantly localises to the nucleus but it is also present in the cytosol [52,53]. 

Most of the mammalian SUMO proteases (SENPs) localise in the nucleus. The mammalian SUMO protease SENP3 and SENP5 localise to the nucleus [54,55]. SENP6 localises throughout the nucleoplasm [56], but catalytically inactive SENP6 concentrates in nuclear foci [57]. In *Arabidopsis*, the equivalent ULP proteases largely localise in the nucleus. Among the various ULPs studied, ESD4 predominantly localises to the nuclear periphery and envelope [58,59]. OTS1 and OTS2 localise to the nucleus, and OTS2 is also found in nuclear foci [5]. Additionally, the recent characterisation of SPF1 and SPF2 showed that they are nuclear as well [60,61].

STUbLs are localised in the nucleus in yeast and mammals to foci formed at sites of DNA breaks and are required for genome integrity [46]. Among the six STUbLs identified in Arabidopsis, only At-STUbL3 is localised in the cytoplasm, while the rest are localised in the nucleus [46]. If elaborated, the STUbLs have different localisation patterns within the nucleus. AtSTUbL1, 3, and 4 localise in the nucleolus while remaining localised in the nucleoplasm [46].

## 6. Localisation of SUMO Components in the Cytoplasm and Plasma Membrane

Studies to date indicate that the E1 enzymes have been found mostly in the nucleus. Evidence suggests that in animal systems, proteins localised in other cellular compartments except the nucleus also become SUMOylated. However, as the E1 enzyme functions primarily in the nucleus, it is necessary that the target protein is either localised in the nucleus or stays in the nucleus for a certain period of time [49]. However, no such phenomenon has been established in plants. The evidence of SUMOylation in the plasma membrane and cytoplasm in plant cells [9,12] indicates additional E1 conjugating enzymes residing in other sub-cellular compartments that need to be investigated.

Transient expression assays suggest that Arabidopsis E2 conjugating enzyme is distributed between the nucleus and cytoplasm; however, when co-expressed with SUMO, it mainly localises to the nucleus, and to nuclear speckles. This SUMO-mediated SCE1 redistribution is dependent on its activity since the SCE1 C94S mutant does not display this capacity [62]. In similar experiments, SCE1 also displays nuclear localisation when co-expressed with the SAE2-UFDCt domain [62]. SCE1 is also observed to interact with FLS2, a membrane protein [10], suggesting that SCE1 subcellular localisation greatly depends on interactions established with other members of the SUMO conjugation machinery.

The SUMO E3 ligases are also localised primarily in the nucleus. However, a second known E3 ligase, MMS21, predominantly localises to the nucleus but it is also present in the cytosol [62,63]. Similar to E2, it would be interesting to analyse if the localisation of any of these ligases is modulated by interactions with other components of the SUMOylation machinery or if there are more SUMO E3 ligases localised in other sub-cellular compartments of the cell.

As mentioned before, most of the SUMO proteases in animal systems are localised in the nucleus; however, certain SENPs have been visualised in the cytoplasm [63,64]. As all SENPs are synthesised in the cytoplasm, SENP-mediated deSUMOylation is likely to occur to some extent in this cellular compartment. DeSI1 is diffusely localised throughout in the cytoplasm and nucleus and DeSI2 concentrates in the cytoplasm in the animals.

In plants, in addition to the nucleus, certain SUMO proteases have been reported to be localised in the cytoplasm and plasma membrane, as studied in Arabidopsis. ELS1/ULP1a, the closest homologue of ESD4, is the only known SUMO protease to date localised predominantly in the cytoplasm [8,65]. The only DeSI currently characterised in *Arabidopsis*; DeSI3a is reported to be localised in the cell membrane [12].

## 7. Distribution of SUMO Machinery Components across Different Organs in Plants

The SUMO machinery targets different proteins simultaneously in response to different stress conditions. The study of the mRNA levels of the SUMO components indicate that SUMO components are expressed throughout the plant [66]. However, the proteins accumulate to different levels in different organs [66]. Figure 3 gives an overview of the protein abundance of the identified SUMO components across different organs in Arabidopsis. When the mRNA profiles of SAE1a, SAE1b, SAE2, SCE1, and SIZ1 were analysed, each was observed to be expressed in all tissues examined, with little variation in their relative levels, indicating that SUMO conjugation is ubiquitously present [66]. *SAE1a* and *SAE1b* encode very similar polypeptides and have comparable expression patterns. The E2, SCE1, appears to be the most highly expressed among all the components studied. However, when the protein levels were studied for these components, it was observed that the levels of SCE1 were particularly high in flowers and siliques, whereas SAE1 levels were slightly elevated in flowers [66]. 

When the expression pattern for *AtSIZ1* was studied using the pSIZ1:GUS reporter construct, it was observed that in 3-day old germinating Arabidopsis seedlings, SIZ1 was expressed in all organs except in parts of the hypocotyl and the basal regions of the cotyledons. A similar expression pattern in the hypocotyl was observed in 3-week-old plants. *SIZ1* was not expressed, at detectable levels, in juvenile leaves and in the basal regions of developing young leaves. However, in developing adult leaves, *SIZ1* expression was detected in leaf blades and petioles. *SIZ1* was strongly expressed in the root system, especially in the primary root and lateral root tips. *SIZ1* expression was also observed in inflorescence stems, sepals, stamen filaments, and stigma, and only low expression levels were detected in anther and vascular tissues of petals. In young developing siliques, *SIZ1* expression was mainly found in the stigma and pedicel, while in adult siliques, expression was present all over the carpel [4]. The mRNA of the second known SUMO E3 ligase MMS21/HPY2 exhibits a differential expression pattern as compared to SIZ1. To study their expression pattern, HPY2pro:GUS transgenic Arabidopsis lines were generated. The studies showed that the expression of *HPY2* is more restricted compared to that of *SIZ1*. *HPY2 promoter* was found to be active in the anther, vein, and hypocotyl cells in the aerial parts. In roots, *HPY2* was expressed relatively strongly in the proliferating cells at the meristem and weak expression was seen in the vascular cells of mature roots [67].

Similar to the SUMO conjugation enzymes, SUMO proteases are also localised in all tissues in the plant at the transcript stage. The highest concentration of the mRNA of the SUMO protease ESD4 was detected in the inflorescence and flowers. However, it was also observed to be expressed at lower concentrations in the seedlings, leaves, shoots, and roots of wild-type plants and was found to be constantly present throughout the 24-hour cycle [59]. An ELS1/ULP1a promoter-GUS fusion showed that ULP1a is expressed ubiquitously in the plant, with a higher accumulation in the vasculature and roots. RT-PCR demonstrated that there is also high expression in flowers and low levels of expression in the siliques and leaves [68].

The GUS (β-Glucuronidase) staining assays show that the SUMO proteases OTS1 and OTS2 have similar expression patterns and are present from the early developmental stages, with high expression levels in vascular tissue in the roots and shoots of seedlings and in the petioles. However, in mature plants, the expression is reduced in the leaves as compared to the seedlings. The proteases were also identified in the flowers and siliques, with OTS2 expression stronger than OTS1; for most other tissues, OTS1 expression was stronger [57]. 

The GUS reporter system for SPF1 expression found ubiquitous expression in 2-day-old seedlings. In 4-day-old seedlings, SPF1 was detected in the hypocotyl, cotyledons, and shoot and root apices of the seedlings. In older seedlings, it is present in newly developing leaves, shoot apex, and root tips [61,62]. SPF1 and SPF2’s strongest gene activity was detected in the reproductive organs, specifically localising to embryo sacs, inflorescences, anthers, and developing seeds [61,62]. The mRNA expression level of SPF1 was highest in inflorescences and cauline leaves, with intermediate expression levels in stems and rosette leaves. However, the transcript level of SPF2 was seen to be the highest in stems, cauline leaves, rosette leaves, and middle-length siliques, and interestingly, no expression of SPF2 was detected in root tissue [62].

## 8. Physiological Effects of SUMOylation in Plants

Although the SUMO system has been well studied in animals, research on the SUMO mechanism in plants is still emerging. Recent research has shown SUMOylation and SUMO proteases in various pathways regulating not only plant growth and development but also in defence responses against various biotic and abiotic stresses. SUMO conjugation has been associated with multiple hormone signalling pathways. This reflects the potential of SUMOylation and SUMO proteases in regulating fundamental processes in plants. SUMO proteases localised in different sub-cellular compartments affect different physiological processes and affect the SUMO status and thereby the functionally of the target proteins within the sub-cellular fraction in which they occur.

## 9. SUMO Conjugation and Deconjugation in the Nucleus

SUMOylation has been found to regulate many factors affecting plant growth and immunity. The SUMO machinery is involved in the regulation of photomorphogenesis through phytochromes and COP1, the master repressor of photomorphogenesis [11,69]. The E3 ligase SIZ1 SUMOylates COP1 in the dark, increasing its transubiquitination activity. Prolonged light exposure reduces the SUMOylation level of COP1, and COP1 mediates the ubiquitination and degradation of SIZ1, thus maintaining homeostasis of COP1 that ensures proper photomorphogenic development [69]. The SUMO protease OTS1 causes deSUMOylation of phytochrome b (PHYB), the photoreceptor for red light [11]. PHYB is SUMOylated in high light, preventing its interaction with the PIFs (Phytochrome Interacting Factors), resulting in the inhibition of hypocotyl elongation. In the dark, OTS1 deSUMOylates PHYB that, at least partly, contributes to hypocotyl elongation [11].

SUMOylation affects plant development and defence through regulating various hormonal pathways. The SUMO protease OTS1 cleaves SUMO from ARF7 (auxin response factor 7), affecting hydropatterning [10]. ARF7 is deSUMOylated by OTS1/2 in the presence of water, causing the transcription of ARF7 target genes and the formation of lateral roots [10]. In the gibberellic acid (GA) pathway, SUMOylation determines the interaction of GA receptor DELLA proteins [5]. The DELLA proteins interact with and control the stability of SLY1 (sleepy1), which forms the Skp, CULLIN, F-box (SCF), and complex of E2 ubiquitin ligases that polyubiquitinate and degrade the DELLA proteins. SLY1 encodes an F-box protein that provides substrate specificity of the SCF complex recognising and binding the DELLA proteins. SUMOylated DELLA proteins are stabilised and therefore accumulate and act to inhibit growth through DNA binding. DeSUMOylation of DELLA proteins by OTS1 allows these proteins to interact with the GA receptors, resulting in their degradation and therefore repression of the DELLA inhibitory pathway, thus allowing plant growth [5]. Similar to the OTSs, DELLAs have been studied to be functional in the roots and aerial parts of the plant [70]. SUMO E3 ligase SIZ1 causes the SUMOylation of SLY1, stabilising and activating it, and is deSUMOylated by a second SUMO protease ESD4 identified in the nucleus [71]. In *esd4-1* mutant plants, there were higher levels of SLY1 protein, due to higher levels of SLY1 SUMOylation resulting in stability and activation of SLY1, degrading more DELLA proteins and resulting in more GA signalling [71].

The involvement of SUMO machinery in salicylic acid (SA) signalling studied to date is through the SUMO proteases OTS1/2 and ESD4. Both the *ots1ots2* and *esd4* mutants exhibit upregulated expression of the SA biosynthetic gene *ICS1* [23,72,73]. Both the SUMO proteases are degraded on treatment with SA that promotes the accumulation of SUMO 1 conjugates. These results indicate that SUMO homeostasis is required for SA biosynthesis in Arabidopsis and elevated levels of SA strongly increase the abundance of SUMO conjugates [23,72,73]. A variety of SUMO related mutants, including *esd4-1, ots1-1, ots2-1, siz1-1*, and *sum1-amiR sum2*, and SUMO overexpression lines show hallmarks of an increased SA response [23,72,73] However, whether the SUMO proteases OTS1/2 and ESD4 contribute together or affect separate pathways has not yet been studied.

The SUMO protease OTS1/2 regulates jasmonic acid (JA) signalling as well. Srivastava et al. (2018) have shown that regardless of intrinsic JA levels, SUMO-conjugated JAZ proteins inhibit the JA receptor Coronatine Insensitive1 (COI1) from mediating non-SUMOylated JAZ degradation. OTS1/2 have been reported to regulate JAZ protein SUMOylation and stability [9]. Necrotroph infection of *Arabidopsis thaliana* promotes SUMO protease degradation, and this increases JAZ SUMOylation and abundance, which in turn inhibits JA signalling [9]. 

SUMO also plays a role in mediating the temperature-dependent trade-off between plant immunity and growth through the regulation of the interaction of PIF4 and SNC1 (Suppressor of *npr1-1* Constitutive 1) [74]. SIZ1 regulates immunity through TPR1 (ToPless-Related 1) as well. SUMO E3 ligase SIZ1 physically interacts with TPR1, facilitating its SUMO modification that represses plant immunity. Hence, SIZ1 functions as a suppressor of plant immunity in Arabidopsis [75]. SIZ1 also regulates the uptake of nutrients including phosphates and heavy metals such as copper [1,76]. AtSIZ1 is majorly involved in Pi starvation responses by controlling a repressor system that attenuates gene expression and developmental responses that are only derepressed (activated) by Pi starvation [1]. Excess copper stimulates the accumulation of SUMO1 conjugates with the help of SIZ1, which regulates the expression of metal transporters *YSL1* (Yellow Stripe-Like 1) and *YSL3*. SIZ1-mediated SUMOylation is involved specifically in copper homeostasis and tolerance in planta [76]. The SUMO protease identified in copper tolerance is OTS1/OTS2 [76]. Under excess copper, OTS1 regulates photosynthetic activity and ROS accumulation. OTS1 regulates copper uptake and distribution through the regulation of the expression of genes responding to high copper when placed on excess copper media [77]. 

SUMOylation is studied to regulate cell division as well as circadian rhythms. The SUMO E3 ligase HPY2/MMS21 is associated mainly with cell division. The SUMO E3 ligase, HPY2, functions as a repressor of endocycle onset in Arabidopsis meristems. Loss of HPY2 results in a premature transition from the mitotic cycle to the endocycle, leading to severe dwarfism with defective meristems. HPY2 is predominantly expressed in the proliferating cells of root meristems and it functions downstream of meristem patterning transcription factors Plethora1 (PLT1) and PLT2 [46]. HPY2 is also involved in regulating cell division and organ size in the model ornamental plant *Petunia multiflora* [78]. However, there is no information yet about the protein that is SUMOylated by HPY2. HPY2 has also been studied to regulate cell proliferation in the root via cell cycle regulation and cytokinin signalling in Arabidopsis [46]. Thus, HPY2 represents a new class of negative regulators of root development involved in cell cycle regulation. HPY2 is also involved in the SUMOylation of Flowering Locus C (FLC), which is a floral repressor and plays an important role in flowering. FLC is stabilised by HPY2 mediated SUMOylation at both the transcriptional and post-translational levels [74]. The SUMO proteases OTS1 and OTS2 regulate circadian clock by controlling the SUMOylation status of CCA1 (Circadian Clock-Associated 1), a novel regulator of key clock proteins, at the end of the night/dawn phase. Imbalances in the SUMOylation of CCA1 could have significant consequences for the growth and health of plants [79]. SUMO protease OTS1/2 has also been characterised in rice as well. In rice, knocking out OTS1/2 promotes drought tolerance; rice OTS1-RNAi lines are much more sensitive to ABA (ABscissic Acid) and survive better in drought conditions, losing less water. OsOTS1 interacts with OsbZIP, a transcription factor that regulates ABA and drought responses [7]. OsOTS1 RNAi lines have been reported to have lower germination success rates; this physiological trait can be a marker of ABA [6]. 

Two SUMO proteases have been recently identified, SUMO Protease related to Fertility 1 (SPF1) and SPF2, which regulate male and female gamete and embryo development. *Spf1* mutants exhibit abnormal floral structures and embryo development, while *spf2* mutants exhibit largely a wild-type phenotype. However, *spf1spf2* double mutants exhibit severe abnormalities in microgametogenesis, megagametogenesis, and embryo development, suggesting that the two genes are functionally redundant. Mutation of *SPF1* and *SPF2* genes also results in the misexpression of generative and embryo-specific genes. SPF1 has been reported to interact with Embryo sac Development Arrest 19 (EDA19) [62,80,81].

Among the different STUbLs identified in Arabidopsis, mutations in *AtSTUbL4* delay flowering, probably due to the impaired degradation of a floral repressor called Cyclin Dof Factor2 (CDF2), because the overexpression of *AtSTUbL4* reduces CDF2 abundance and accelerates flowering [45].

## 10. SUMO Conjugation and Deconjugation in Other Cellular Compartments

Not many components of SUMO machinery have been identified to date that function in other cellular compartments besides the nucleus. Although SUMO E3 ligase HPY2/MMS21 have been reported to be localised in the cytoplasm as well as nucleus [52,82], the reports found to date discuss its nuclear localisation. However, a few SUMO proteases have been identified that function in the plasma membrane and cytoplasm in the cell.

DeSI3a is a plasma membrane-bound SUMO protease that plays a role in PAMP (Pathogen-Associated Molecular Patterns) detection. Upon detection of flagellin, FLS2 is SUMOylated, triggering the release of BIK1 (Botrytis-Induced Kinase 1), a cytoplasmic kinase, resulting in downstream signalling in innate immunity. When flagellin is perceived, DesI3a is degraded, which enhances FLS2 SUMOylation, triggering BIK1 dissociation and downstream intracellular immune signalling [12]. Another SUMO protease, ULP1a, localised in the cytoplasm, plays a role in the response to salt stress by regulating the brassinosteroid signalling pathway. It is reported that the SUMOylation of BZR1, a key transcription factor for BR (brassinosteroid) signalling, provides a conduit for environmental influences to modulate growth during stress. SUMOylation stabilises BZR1 in the nucleus by inhibiting its interaction with BIN2 kinase. During salt stress, Arabidopsis plants arrest growth through the deSUMOylation of BZR1 in the cytoplasm by promoting the accumulation of the BZR1 targeting SUMO protease, ULP1a. ULP1a mutants are salt-tolerant and insensitive to the BR inhibitor brassinazole. BR treatment stimulates ULP1a degradation, allowing SUMOylated BZR1 to accumulate and promote growth [8].

## 11. Future Perspective

Although lately there has been much focus on the SUMO machinery, there are still many questions that need to be answered. Studies need to be carried out to determine the factors that provide specificity to the process. Being a highly dynamic process involved in a multitude of processes, there must be strict measures that regulate the association of SUMO with the target protein. Whether this regulation resides on specific components of the SUMO machinery, such as the E3 ligases, SUMO proteases, or other factors, still needs to be determined. Moreover, very few SUMO conjugation enzymes and E3 ligases have been identified. Whether there are more enzymes to add to the list needs to be ascertained. The identification of additional SUMO proteases can be considered as a hint at the existence of more SUMO components that are yet to be identified.

To date, the SUMO activation enzymes identified have been found only in the nucleus. Although the E3 SUMO ligase HPY2 has been found to be localised in the cytoplasm along with nucleus, the reports only discuss its nuclear localisation. As SUMOylation of targets has been reported in other cellular components and novel SUMO proteases have been identified as localising in other cellular components besides the nucleus, the existence of additional SUMO components in these compartments may be possible and is yet to be discovered. Additionally, SUMOylation may affect processes in other cellular compartments like mitochondria or chloroplasts.

There is also the need to extend the study of SUMOylation beyond the model system of Arabidopsis into crops. SUMOylation is critical for stress responses [2,17,40,83]. This leads to the hypothesis that crops may have a greater possibility of SUMOylation regulating their physiological responses due to the selection pressures that have been applied to domesticated plants and breeding for tolerance [84,85]. However, there has been limited research in crops. Many components of the SUMO system have been identified by bioinformatics techniques for rice, wheat, maize, and soybean [86,87], and the maize SUMO system has been reconstituted [84]. From the limited research into crop SUMO proteases, evidence indicates that SUMO is vital to crop productivity under stress. Further studies into a wider variety of crops and a detailed study of all the SUMO components will likely provide greater insights into their important role in stress survival and may prove a useful target for breeders.

Since its discovery over 20 years ago, the roles of SUMO in many different processes in plants have been uncovered. The role of SUMO in many vital aspects of plant biology, from hormonal signalling pathways to abiotic and biotic responses, has demonstrated that SUMO is a critical post-translational modification in plants. With the large number of biological processes existing in plants and so few SUMO components identified, it is likely that more SUMO components will be identified, with roles in numerous other pathways and in other components of the plant cell.

## Figures and Tables

**Figure 1 plants-09-01788-f001:**
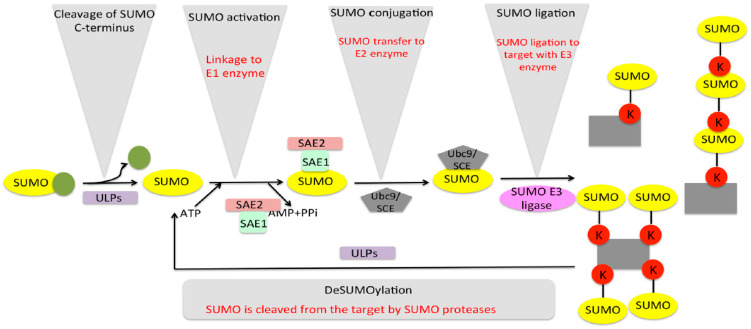
SUMO cycle of activation, conjugation, and deconjugation. The SUMO cycle starts with maturation of immature SUMO by cleaving off C-terminus by SUMO isopeptidase. Mature SUMO is then activated by E1 enzymes in an ATP-dependent manner. The SUMO then conjugates to the target using conjugating and ligating enzymes E2 and E3. The target can be SUMOylated aa more than one SUMO proteases, the process termed as deSUMOylation, and the free SUMO is available for another SUMO cycle.

**Figure 2 plants-09-01788-f002:**
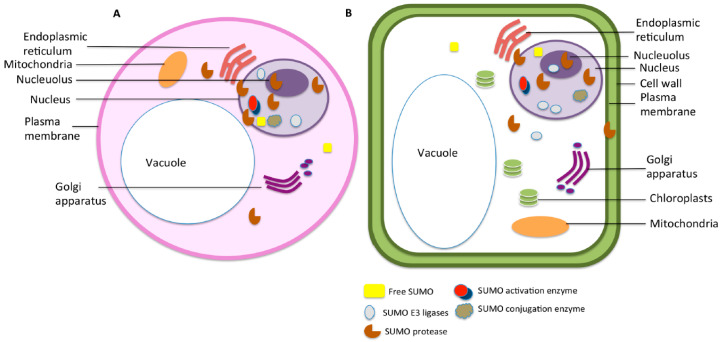
Subcellular distribution of components of SUMO machinery in (**A**) animal cell and (**B**) plant cell.

**Figure 3 plants-09-01788-f003:**
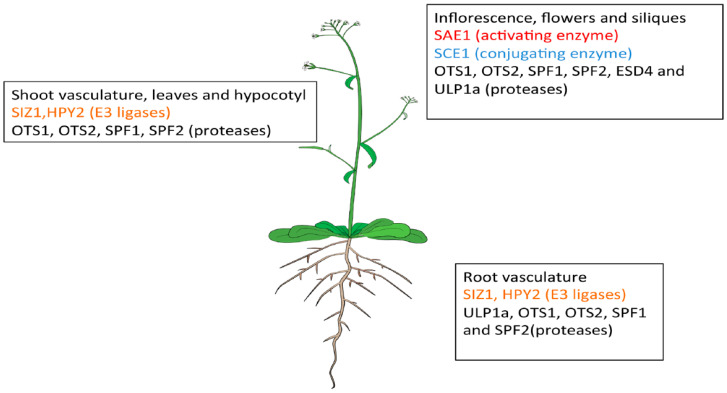
Distribution of SUMO components in different organs in *Arabidopsis thaliana*. The figure concentrates on the expression of different components at the adult stage. The figure depicts the regions where the protein is particularly expressed at higher concentrations.

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
