# Peer review of "An Insight into the Factors Influencing Specificity of the SUMO System in Plants"

_plants, 2020, doi:10.3390/plants9121788_

Round 1

Reviewer 1 Report

Srivastava and Sadanandom provide an overview of the literature concerning SUMO in plants. The review is well written and easy to read. They first describe the proteins known to be at play in the plants SUMO pathway, with a special emphasis on their localization, both intracellular and across the different organs. They then describe the known physiological effects of SUMOylation in plants covering the various hormonal pathways (gibberellic acid, salicylic acid and jasmonic acid signaling). They also cover how SUMO pathway affects circadian rythms and drought tolerance, which has broader socio-economic implications.

I only have minor comments. The Figure 1 could be improved: the police size is too small and almost unreadable at this scale when printed.  Line 312, the meaning of the acronym “ABA” should be indicated.

Author Response

Reviewer 1:

Srivastava and Sadanandom provide an overview of the literature concerning SUMO in plants. The review is well written and easy to read. They first describe the proteins known to be at play in the plants SUMO pathway, with a special emphasis on their localization, both intracellular and across the different organs. They then describe the known physiological effects of SUMOylation in plants covering the various hormonal pathways (gibberellic acid, salicylic acid and jasmonic acid signaling). They also cover how SUMO pathway affects circadian rythms and drought tolerance, which has broader socio-economic implications.

I only have minor comments.

The Figure 1 could be improved: the police size is too small and almost unreadable at this scale when printed. 

Response: We thank the reviewer for the valuable comments. We have generated a new Figure 1 taking into account the reviewer’s comments on the font size to ensure its easily readable

Line 312, the meaning of the acronym “ABA” should be indicated.

Response: We thank the reviewer for pointing this out. The meaning of the acronym has now been included in the revised manuscript (Line 395).

Reviewer 2 Report

The small ubiquitin-related modifier (SUMO) system is an evolutionarily conserved system that functions by becoming covalently attached to SUMO themselves and other proteins as post-translational modifications. As described in the review manuscript by Srivastava M. and Sadanandom, the SUMO system in plants can regulate multiple cellular pathways, including plant growth, development and defense responses against various stresses. Also, the SUMO system in plants are more complex than those in budding yeast S. cerevisiae and animals. For example, the Arabidopsis genomes encodes 8 SUMO isoforms, whereas budding yeast S. cerevisiae and human only possess one and four isoforms, respectively. It is intriguing to speculate that different SUMO isoforms or components could be expressed in condition-, time- or space-specific manners. Accordingly, in this review, the authors have highlighted the functional impacts of the sub-cellular localization of the SUMO components in higher plants.

The manuscript is well written and provides valuable insights about the SUMO system in plants. 

I would suggest four minor modifications:

  1. Several references [#93-#97 in page 9] are missing in the review manuscript. The authors should discuss the number of SUMO isoforms in the genomes of other (crop) plants ?

  1. The SUMO-targeted ubiquitin ligases (STUbL) have important role in SUMO-targeted degradation and SUMO-dependent signaling, including the Slx5/Slx8 heterodimer in budding yeast and and the RING-domain-containing ubiquitin E3 ligase, RNF4 (also known as SNURF) in animal cells. The authors have to discuss whether there are STUbL orthologs in plant and provide an introduction about the roles and subcellular localization of plant STUbL

  1. Poly-SUMO chains are known to have important cellular functions in yeast and animal cells. In the manuscript's current form, it is unclear if poly-SUMO chains exist in plants and whether the poly-SUMO chains in plants are homo- or hetero-SUMO chains. Also, do poly-SUMO chains preferentially accumulate in specific sub-cellular location?

  1. A hallmark of the SUMO system is that the non-covalent interactions between SUMO or SUMO conjugates with their binding proteins is mediated by short amino acid consensus sequences termed SUMO-interacting motifs (SIMs). The authors need to provide an introduction about SIMs in plants. Do plants SIMs regulate the subcellular localization of SUMO components in plants ?

Author Response

Reviewer 2

The small ubiquitin-related modifier (SUMO) system is an evolutionarily conserved system that functions by becoming covalently attached to SUMO themselves and other proteins as post-translational modifications. As described in the review manuscript by Srivastava M. and Sadanandom, the SUMO system in plants can regulate multiple cellular pathways, including plant growth, development and defense responses against various stresses. Also, the SUMO system in plants are more complex than those in budding yeast S. cerevisiae and animals. For example, the Arabidopsis genomes encodes 8 SUMO isoforms, whereas budding yeast S. cerevisiae and human only possess one and four isoforms, respectively. It is intriguing to speculate that different SUMO isoforms or components could be expressed in condition-, time- or space-specific manners. Accordingly, in this review, the authors have highlighted the functional impacts of the sub-cellular localization of the SUMO components in higher plants.

The manuscript is well written and provides valuable insights about the SUMO system in plants. 

I would suggest four minor modifications:

  1. Several references [#93-#97 in page 9] are missing in the review manuscript. The authors should discuss the number of SUMO isoforms in the genomes of other (crop) plants ?

Response: We thank the reviewer for pointing this out. The reference numbering has been corrected. We have also discussed about the number of SUMO isoforms in the genomes of other crop plants in the revised manuscript (Line 69-72).

  1. The SUMO-targeted ubiquitin ligases (STUbL) have important role in SUMO-targeted degradation and SUMO-dependent signaling, including the Slx5/Slx8 heterodimer in budding yeast and and the RING-domain-containing ubiquitin E3 ligase, RNF4 (also known as SNURF) in animal cells. The authors have to discuss whether there are STUbL orthologs in plant and provide an introduction about the roles and subcellular localization of plant STUbL.

Response: A section has been added in the present manuscript explaining about STUbL orthologs in plants (Lines 163-177).

  1. Poly-SUMO chains are known to have important cellular functions in yeast and animal cells. In the manuscript's current form, it is unclear if poly-SUMO chains exist in plants and whether the poly-SUMO chains in plants are homo- or hetero-SUMO chains. Also, do poly-SUMO chains preferentially accumulate in specific sub-cellular location?

Response: We thank the reviewer for bringing up the question. It has been observed that in Arabidopsis, the capacity to build polySUMO chains is restricted to SUMO 1/2 isoforms, while SUMO3 and 5 are mainly conjugated with monomers (Chosed et. al., 2006; Colby et, al., 2006). Although there is information about the existence of polySUMO chains in plants, not much information is available about its accumulation in any particular sub-cellular location. We also do not have any information whether polySUMO chains are homo or hetero SUMO chains in plants. Hence we are reluctant to pursue this aspect in the current revision.

  1. A hallmark of the SUMO system is that the non-covalent interactions between SUMO or SUMO conjugates with their binding proteins is mediated by short amino acid consensus sequences termed SUMO-interacting motifs (SIMs). The authors need to provide an introduction about SIMs in plants. Do plants SIMs regulate the subcellular localization of SUMO components in plants?

Response: We thank the reviewer for raising the point. A section has been added in the present manuscript that discusses SUMO-interacting motifs (SIMs) (Line 142-149).

            An increasing number of proteins have been shown to bind SUMO or SUMO chains non-covalently via SIMs. Protein SUMOylation provides an interaction platform for the recruitment of SIM-containing effector proteins. In addition to SIMs mediating the effects of SUMOylation, a growing number of proteins have been identified for which SUMOylation is dependent on the presence of SIMs in the substrate. So although SIMs have been shown to have important roles in facilitating SUMO conjugation, no evidence about SIMs regulating subcellular localisation of SUMO components have been studied in plants yet.

References:

  1. Colby, T.; Matthai, A.; Boeckelmann, A.; Stuible, H.P. SUMO-conjugating and SUMO-deconjugating enzymes from Arabidopsis. Plant Physiol. 2006, 142(1), 318-332.
  2. Chosed, R.; Mukherjee, S.; Lois, L.M.; Orth, K. Evolution of a signaling system that incorporates both redundancy and diversity: Arabidopsis SUMOylation. Biochem J., 2006, 398, 521-529.